# Triplon band splitting and topologically protected edge states in the dimerized antiferromagnet

Kazuhiro Nawa [1], Kimihiko Tanaka[2], Nobuyuki Kurita[2], Taku J. Sato[1], Haruki Sugiyama[3], Hidehiro Uekusa[3], Seiko Ohira-Kawamura[4], Kenji Nakajima[4] & Hidekazu Tanaka[2]

Search for topological materials has been actively promoted in the field of condensed matter physics for their potential application in energy-efficient information transmission and processing. Recent studies have revealed that topologically invariant states, such as edge states in topological insulators, can emerge not only in a fermionic electron system but also in a bosonic system, enabling nondissipative propagation of quasiparticles. Here we report the topologically nontrivial triplon bands measured by inelastic neutron scattering on the spin-1/2 two-dimensional dimerized antiferromagnet $Ba_2CuSi_2O_6Cl_2$. The excitation spectrum exhibits two triplon bands that are clearly separated by a band gap due to a small alternation in interdimer exchange interaction, consistent with a refined crystal structure. By analytically modeling the triplon dispersion, we show that $Ba_2CuSi_2O_6Cl_2$ is the first bosonic realization of the coupled Su-Schrieffer-Heeger model, where the presence of topologically protected edge states is prompted by a bipartite nature of the lattice.

[1] Institute of Multidisciplinary Research for Advanced Materials, Tohoku University, 2-1-1 Katahira, Sendai 980-8577, Japan. [2] Department of Physics, Tokyo Institute of Technology, Meguro-ku, Tokyo 152-8551, Japan. [3] Department of Chemistry, Tokyo Institute of Technology, Meguro-ku, Tokyo 152-8551, Japan. [4] Materials and Life Science Division, J-PARC Center, Tokai, Ibaraki 319-1195, Japan. Correspondence and requests for materials should be addressed to K.N. (email: knawa@tohoku.ac.jp) or to H.T. (email: tanaka@lee.phys.titech.ac.jp)

The discoveries of quantum Hall effects[1] and topological insulators[2] have shed light on topologically protected gapless edge states that exist between phases with different topological characters[3,4]. The edge states preserve dissipationless particle/quasiparticle flow that could be useful for future applications in energy-efficient information transmission and processing. Recently, the concept of the edge states have been extended to other systems, such as ultracold atom systems in optical lattices[5–7], and even to systems with bosonic quasiparticles, such as photonic crystals[8,9], phonons[10], and magnons[11–15] in solids. In electron systems, the topological characters are classified by the total topological invariant of the occupied bands, which is associated with quantized Hall conductance[3,4,16]. In contrast, for magnetic insulators, the electric conductance is zero by definition. Instead, thermally excited bosonic quasiparticles convey a heat transport, and hence thermal conductance is supposedly the key transport property that reflects the topological nature of the underlying quasiparticle dispersion relations[12,17,18]. Detailed knowledge on dispersion relations of the excited states are, therefore, necessary to explore and design the magnetic insulators with bosonic topological bands.

A dimerized magnet, which has well-defined bosonic excitations called triplons, is a good starting point for realizing the bosonic topological bands[19–21]. Because of the dominant antiferromagnetic intradimer interactions, triplons are locally formed with a finite energy gap from the singlet ground state at each dimer. The transverse and longitudinal terms of the interdimer exchange interactions induce the hopping and repulsion of the triplons, respectively[19–21]. Thus, the triplet excitations can be modeled as interacting bosonic quasiparticles.

One of the advantages of studying a dimerized magnet is that the triplon bands can be easily deformed by applying a magnetic field or hydrostatic pressure. If the deformation is so large that a triplet excitation energy becomes zero, a quantum phase transition will occur[20–23]. For instance, with an increasing magnetic field, an $S^z = +1$ branch of triplons will undergo Bose-Einstein condensation (BEC) if the kinetic energy of triplons is more dominant than repulsive interactions[24–30], while a Wigner crystal of localized triplons will be realized in the opposite case[31–34].

Quite interestingly, recent theoretical advancement has revealed that topological triplons can be realized in a certain dimerized magnet[35–37]. For instance, the $S^z = +1$, 0, and −1 branches of triplons are predicted to become topologically non-trivial under an applied magnetic field in $SrCu_2(BO_3)_2$ owing to interdimer Dzyaloshinskii–Moriya interactions which yield complex hopping amplitudes[35]. This prediction is substantiated by the result of inelastic neutron scattering experiments combined with detailed calculations of the winding (Chern) numbers and the edge-state spectrum to a very high accuracy[38]. The transition between topologically trivial and non-trivial phases can be tuned by controlling the magnitude or direction of a magnetic field[35,39]. At the critical magnetic field, where the transition between the two phases occurs, the formation of a spin-1 Dirac cone is expected[35].

Recently, a new two-dimensional (2D) quantum dimer compound $Ba_2CuSi_2O_6Cl_2$ has been discovered[40]. This compound crystallizes in an orthorhombic layered structure with each layer composed of antiferromagnetically coupled dimers. The space group was originally reported as $Cmce$[40], whereas a slight distortion to $Cmc2_1$ is confirmed in the present work, as detailed later, allowing an alternation in the interdimer exchange interactions along the $a$-axis. Figure 1a illustrates the 2D exchange network in $Ba_2CuSi_2O_6Cl_2$. A pair of the nearest-neighbor Cu atoms that align almost parallel to the $c$-axis forms an antiferromagnetic dimer via the exchange coupling $J$ as denoted by the black solid line in Fig. 1. These dimers are coupled via interdimer exchange couplings $J_{ij}^\alpha$ and $J_{ij}^{\alpha'}$ with $i, j = 1, 2$ and $\alpha =$ a, b, forming a 2D exchange network in the $ab$ plane. In fact, the magnetic properties of $Ba_2CuSi_2O_6Cl_2$ are well characterized by a spin-1/2 quasi-2D dimer system[40]. The magnetization curve is excellently reproduced using the exact diagonalization calculation based on the 2D coupled dimer model, indicative of the strongly two-dimensional characters in the exchange network. Under the assumptions of $J_p \equiv J_{11}^\alpha = J_{11}^{\alpha'} = J_{22}^\alpha = J_{22}^{\alpha'}$ and $J_d \equiv J_{12}^\alpha = J_{12}^{\alpha'} = J_{21}^\alpha = J_{21}^{\alpha'}$ ($\alpha =$ a, b), the exchange constants were estimated as $J = 2.42$ meV, $J_p = 0.03$ meV, $J_d = 0.34$ meV from the magnetization curve and density functional theory calculations[40]. In addition, the magnetic anisotropy should be very small since magnetic susceptibilities and entire magnetization curves for two different field orientations coincide almost perfectly with each other after being normalized by the $g$-factors.

In this work, we investigated the triplon band dispersion and its topological nature in the 2D quantum dimer compound $Ba_2CuSi_2O_6Cl_2$ using inelastic neutron scattering, as well as model analysis. The main finding of this study is the gap between two triplon bands, as shown in Fig. 2. As we discuss later, this result contradicts the previously reported crystal structure of $Cmce$, which cannot host two triplon bands nor give rise to the gap between them. With the renewed space group $Cmc2_1$, we show that the gap is indeed topologically protected, and hence hosts emergent edge states for the bond-alternation direction. This compound is the first realization of the bosonic analog of the 2D coupled Su-Schrieffer-Heeger (SSH) model, a prototypical model for a one-dimensional topological insulator.

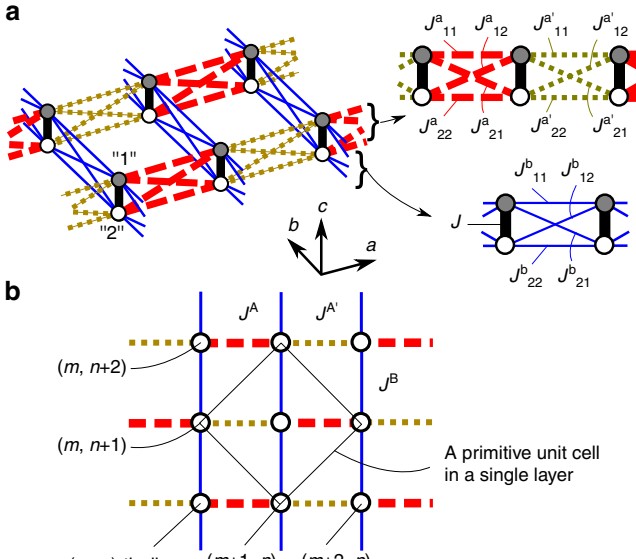

**Fig. 1** Intradimer and interdimer couplings in $Ba_2CuSi_2O_6Cl_2$. **a** Possible interactions between Cu atoms. Shaded and open circles indicate two symmetrically different Cu atoms. The intradimer coupling formed by the two Cu atoms are represented by thick black lines, and the interdimer couplings are represented by dashed red, dotted yellow, and thin blue lines. **b** Effective couplings between dimers expected from the crystal symmetry $Cmc2_1$. Dashed, dotted, and solid lines represent hopping amplitudes denoted by $J^A$, $J^{A'}$, and $J^B$, respectively (see text for definition)

## Results

**Intradimer and interdimer couplings in $Ba_2CuSi_2O_6Cl_2$.** First, we revisit the crystal structure of $Ba_2CuSi_2O_6Cl_2$ using single-crystal X-ray diffraction (XRD) technique. The details of the experiment and the refined structure are described in the

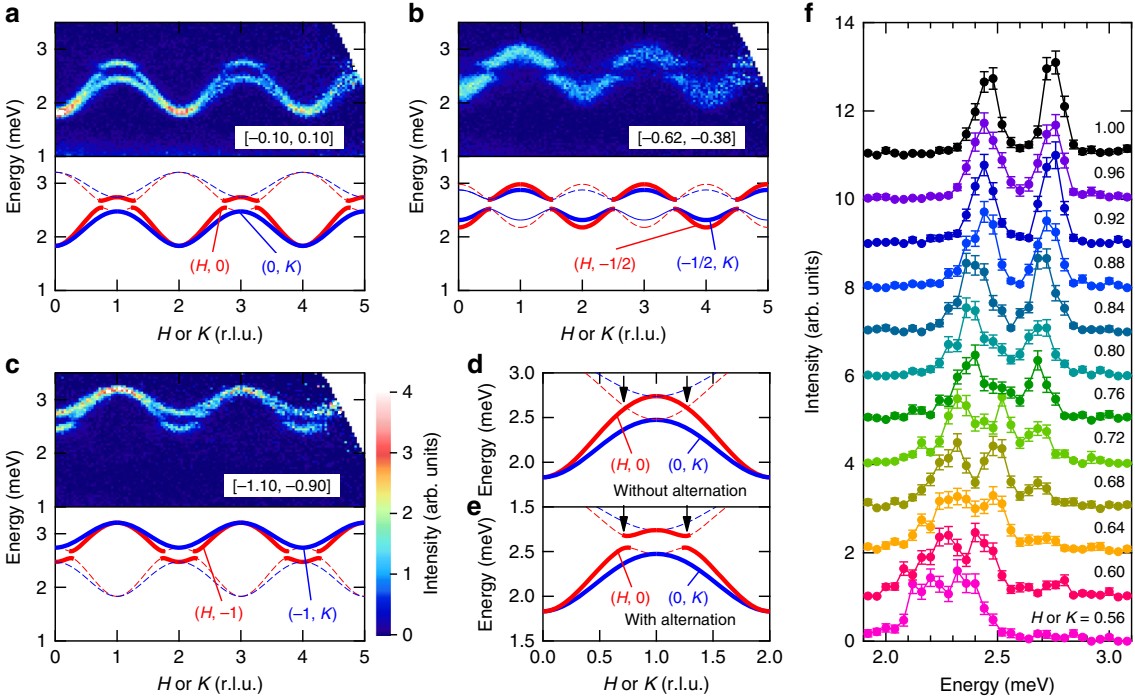

**Fig. 2** Excitation spectra along $H$ or $K$ directions. **a** Energy-momentum maps of the scattering intensities along the $(H, 0)$ direction (shown above) compared with the dispersion relations of Eq. (3) (below). The common color scale is used for Fig. 2a–c. Intensity is integrated for the $K$ range shown in the figure ($\Delta K = \pm 0.10$) and the entire observed $L$ range. The branch along the $(0, K)$ direction is observed, as well as that along the $(H, 0)$ direction because of crystallographic domains. **b, c** The same maps along $(H, -1/2)$ and $(H, -1)$ directions and the corresponding dispersion relations. The same color scale is used for all the maps. The presented data are collected at 0.3 K with an incident energy $E_i$ of 5.92 meV. Thick solid and thin dashed curves indicate triplon bands with a large and small structure factor, respectively. **d, e** Expected triplon bands without and with an alternation in interdimer interactions along the $(H, 0)$ and $(0, K)$ directions. Arrows indicate a crossing point of two triplon bands when the alternation is absent. **f** Intensity plotted as a function of energy transfer $E$ at selected reciprocal space points $(H, 0)$ and $(0, K)$ in Fig. 2a. Error bars represent standard deviation. Source data are provided as a Source Data file

Methods section and Supplementary Note 1, respectively. The space group is renewed from $Cmce$ to $Cmc2_1$; the $a$-glide plane is lost in the new structure. Figure 1a illustrates intradimer and interdimer interactions expected from the crystal symmetry of $Cmc2_1$. Although the absence of the twofold rotation results in the symmetrically inequivalent positions of the two nearest-neighbor Cu atoms, all of the intradimer interactions remain identical. On the other hand, the lack of the glide symmetry allows an alternation of interdimer interactions along the $a$-axis, while those along the $b$-axis remain uniform. Finally, as shown in Fig. 1b, three different hopping amplitudes can be present: $J^A$, $J^{A'}$, and $J^B$, representing $\frac{1}{4}(J^a_{11} + J^a_{22} - J^a_{12} - J^a_{21})$, $\frac{1}{4}(J^{a'}_{11} + J^{a'}_{22} - J^{a'}_{12} - J^{a'}_{21})$, and $\frac{1}{4}(J^b_{11} + J^b_{22} - J^b_{12} - J^b_{21})$, respectively (see Supplementary Note 3 for details).

**Inelastic neutron scattering spectra**. Next, we discuss inelastic neutron scattering intensities sliced along the $H$ (or $K$) direction, which are shown as color contour maps in Fig. 2a–c. Intensity is integrated over the observed $L$ range to obtain good statistics. Note that single crystals used in the experiments include mixed domains where the $a$-axis of a single domain and the $b$-axis of another domain coexist along the same edge. As a result, these two triplon bands can be simultaneously observed when measured along both $H$ and $K$ directions. In addition, the bands exhibit the minimum energy at $(2m, 2n, 0)$ ($m$, $n$: integer), indicating that the triplon propagation is in-phase. Hence, the three hopping amplitudes are all negative, indicating the dominant antiferromagnetic diagonal interactions $J^\alpha_{12}$, $J^\alpha_{21}$, $J^{\alpha'}_{12}$, and $J^{\alpha'}_{21}$, which is consistent with the results of DFT calculations[40].

The contour maps of inelastic neutron scattering sliced along the $L$ direction are shown in Fig. 3. Figure 3a, b represent integrated intensities around $(H, K) = (2, 0)$ [and $(0, 2)$ from different domains] and $(1, 0)$ [and $(0, 1)$], respectively. The excitations along $L$ is dispersionless, irrespective of $H$ and $K$, attesting to the good two-dimensionality in the dimer network. In addition, integrated intensities are modulated along $L$, as should be the case for antiferromagnetically coupled dimers along the $c$-axis[28]. Figure 3c, d show energy-integrated intensities from Fig. 3a, b, respectively. The intensity of perfectly aligned antiferromagnetic dimers can be described by a dimer structure factor $I(Q, \omega) \sim |f(Q)|^2[1 - \cos(\mathbf{Q} \cdot \mathbf{r}_d)]$, where $f(Q)$ and $\mathbf{r}_d$ indicate a form factor of $Cu^{2+}$ and a vector representing intradimer separation, respectively. In $Ba_2CuSi_2O_6Cl_2$, there are four types of dimers with slightly different orientations, and hence the dimer structure factor should be corrected for this dimer misalignment. However, since their canting angle of 0.9° from the $c$-axis is very small, we approximate that all the four dimers are aligned along the $c$-axis. As shown in Fig. 3c, the fit to this equation yields an $\mathbf{r}_d$ of $0.150(1)\mathbf{c}$, which is consistent with $0.148(1)\mathbf{c}$ obtained from the crystal structure. The modulation along $L$ does not depend on $H$ and $K$, further supporting the approximation of the equivalent dimers.

What is not expected for a simple dimer antiferromagnet is the decrease in intensity observed at 2.6 meV (Figs. 2 and 3b), which is almost independent of the scattering wave vector, as shown in Fig. 4 by contour maps of intensities sliced at a constant energy. Apparently, the energy slice at 2.60 meV (Fig. 4b) exhibits much weaker intensities than those at 2.48 (Fig. 4a) and 2.72 meV (Fig. 4c). Note that this intensity decrease at 2.6 meV is not due to

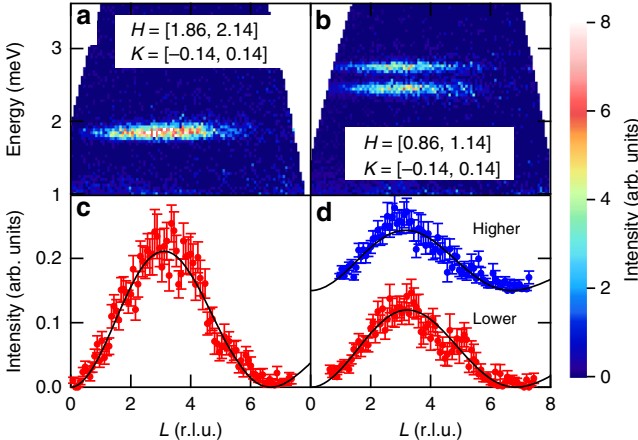

**Fig. 3** Excitation spectra along $L$ directions. **a, b** Color contour maps of scattering intensities along the $L$ ($c^*$) direction. The common color scale is used. Integrated $H$ and $K$ ranges are shown in the figures. **c, d** Integrated intensity plotted against $L$ together with a fit using the dimer structure factor represented by a solid black curve. Red circles in Fig. 3c are obtained by integrating the intensities shown in Fig. 3a between 1.66 and 2.06 meV. Red and blue circles in Fig. 3d correspond to integrated intensities between 2.22 and 2.58 meV and between 2.62 and 2.98 meV from Fig. 3b, respectively. The intensity of the higher branch is shifted upward for clarity. Error bars represent standard deviation. Source data are provided as a Source Data file

an extrinsic effect, because it is unchanged under different measuring conditions (different incident energy $E_i$ and temperature, see Supplementary Fig. 2). The detailed $Q$ dependence of the triplon bands is shown in Fig. 2f, which represents $Q$ slices of Fig. 2a as functions of energy transfer $E$. At small $H$ and $K$, two peaks along the $H$ and $K$ directions overlap. At $H$ and $K$ of 0.68 r. l.u., they start to move apart and form two well-separated peaks. The right peak at 2.50 meV decreases and disappears above 0.80 r. l.u., while the left peak becomes more prominent with increasing $H$ and $K$. Above 0.68 r.l.u., the new peak emerges at 2.68 meV and grows in intensity as $H$ or $K$ increases toward 1.00 r.l.u.

The coexistence of three modes between 0.68 and 0.80 r.l.u. strongly indicates the presence of two triplon bands, which is not allowed if the interdimer interactions along both $a$-axes and $b$-axes are uniform. Starting from the uniform case, we will introduce the bond alternation to explain this phenomenon. Note that the triplon bands are degenerate since the crystallographic unit cell includes eight dimers, which are connected by a mirror symmetry with respect to the $bc$-plane, centering symmetry, and twofold screw symmetry along the $c$-axis. For a simplicity, we will not take into account the fourfold degeneracy caused by the latter two symmetries and instead focus on two dimers in a primitive unit cell with a single layer, as shown in Fig. 1b. In the uniformly interacting case, only one continuous triplon band is detectable, whereas the structure factor of the other is almost zero. The dispersion branches with strong and weak scattering intensities along the $(H, 0)$ and $(0, K)$ directions are depicted as solid and dashed curves in Fig. 2d, respectively. The high-energy band is not observable, because of a very small structure factor resulting from the two almost parallel dimers. In other words, the dimer orientations are so close to each other that triplet excitations cannot be distinguished from those expected from a hypothetical unit cell containing only one dimer. The presence of a triplon band with very weak intensities was also reported in TlCuCl$_3$[26].

When the bond alternation along the $a$-direction is introduced, a band inversion induces a gap between the low-energy and high-energy bands, as shown in Fig. 2e. If the alternation is very small,

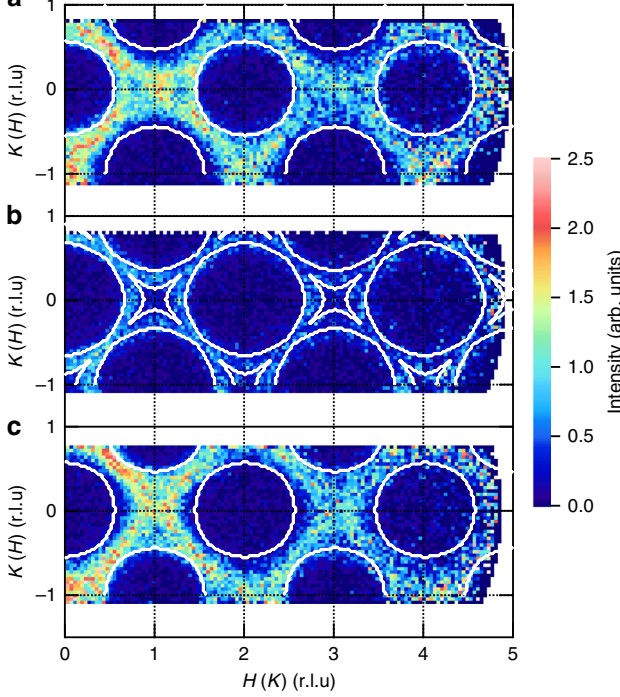

**Fig. 4** Scattering intensities sliced with the constant energy. Intensity is given at **a** 2.48, **b** 2.60, and **c** 2.72 meV for each ($H$, $K$) or ($K$, $H$) position after integrated for $\Delta E = \pm 0.06$ meV and all measured $L$ values. The common color scale is used. White solid curves indicate boundaries between zero and nonzero intensity regions expected from the dispersion relation. Source data are provided as a Source Data file

the structure factor becomes very close to that represented in Fig. 2d. Thus, the intensities of the low-energy and high-energy bands greatly vary around a crossing point of 0.74 r.l.u., indicated by arrows in Fig. 2d, e; the intensities of the high-energy band significantly increases above the crossing point, while those of the low-energy band become undetectable. Even at different $H$ and $K$, the band crossing occurs at the same energy, $J$, since the two dimers are symmetrically equivalent. Consequently, the wavevector-independent gap centered at $J$ appears between two triplon bands. Note that the alternation is only allowed along the $a$-axis owing to the symmetry. Therefore, the observed triplon bands can be labeled to be along $H$ or $K$, as denoted in Fig. 2a–c.

## Discussion

For this purpose, a bond-operator approach[41] is applied to the 2D dimer model represented in Fig. 1a. Triplon bond operators representing a singlet state and triplet states are defined as

$s_{mn}^\dagger|0\rangle = \frac{1}{\sqrt{2}}(|\uparrow\rangle_{mn1}|\downarrow\rangle_{mn2} - |\downarrow\rangle_{mn1}|\uparrow\rangle_{mn2})$,

$t_{xmn}^\dagger|0\rangle = -\frac{1}{\sqrt{2}}(|\uparrow\rangle_{mn1}|\uparrow\rangle_{mn2} - |\downarrow\rangle_{mn1}|\downarrow\rangle_{mn2})$,

$t_{ymn}^\dagger|0\rangle = \frac{i}{\sqrt{2}}(|\uparrow\rangle_{mn1}|\uparrow\rangle_{mn2} + |\downarrow\rangle_{mn1}|\downarrow\rangle_{mn2})$, and

$t_{zmn}^\dagger|0\rangle = \frac{1}{\sqrt{2}}(|\uparrow\rangle_{mn1}|\downarrow\rangle_{mn2} + |\downarrow\rangle_{mn1}|\uparrow\rangle_{mn2})$, respectively, where $m$ and $n$ are labels used to distinguish dimers, and 1 and 2 indicate two Cu atoms in a single dimer. The above definition leads to interacting hard-core bosons characterized by hopping amplitudes $J^A$, $J^{A'}$, and $J^B$, as depicted in Fig. 1b. The detailed calculations are described in the Supplementary Note 3. A $k$-dependent form of the Hamiltonian is obtained by Fourier transformation as

$$\mathcal{H} \sim \frac{1}{2}\sum_{\mathbf{k}}\sum_\alpha \mathcal{T}^\dagger \mathcal{M}_{\mathbf{k}}\mathcal{T}, \qquad (1)$$

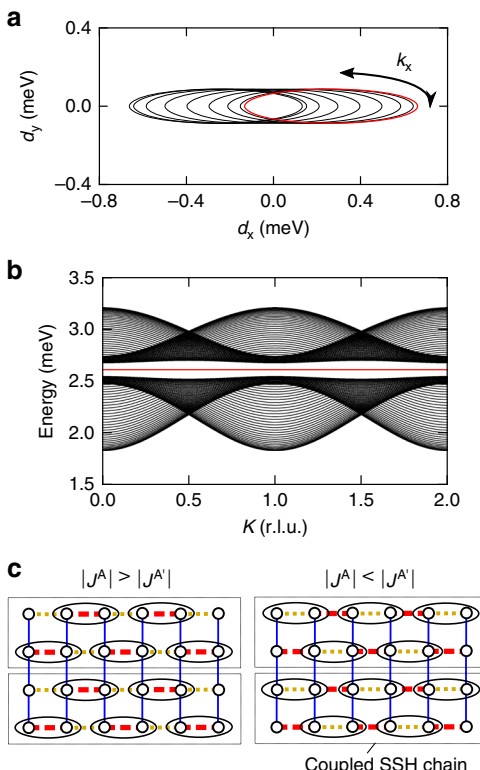

**Fig. 5** Schematic view of expected pseudomagnetic field and edge states. **a** Distribution of a pseudomagnetic field induced on the triplon bands. Each solid curve represents a pseudomagnetic field with a fixed $k_y$ (see Eq. (5) for its definition). **b** Energy spectrum along the $K$ direction. The calculation is based on the model presented in Fig. 1b, in which open boundary conditions with 200 sites (100 for each sublattice) are imposed along the $a$ direction while periodic boundary conditions are set along the $b$ direction. Half of the all modes are shown, and edge states are stressed by a red line for clarity. **c** The interdimer network in Ba$_2$CuSi$_2$O$_6$Cl$_2$ shown as SSH chains coupled by interchain hoppings. The same symbols and lines as those in Fig. 1 are used. An ellipsoid and a dashed rectangle represent a pair of dimers with a larger hopping amplitudes and a single coupled SSH chain, respectively

where

$$\mathcal{T} = \begin{pmatrix} t^1_{\alpha\mathbf{k}} \\ t^2_{\alpha\mathbf{k}} \\ t^{\dagger,1}_{\alpha(-\mathbf{k})} \\ t^{\dagger,2}_{\alpha(-\mathbf{k})} \end{pmatrix}, \mathcal{M}_{\mathbf{k}} = \begin{pmatrix} J & \Lambda_{\mathbf{k}} & 0 & \Lambda_{\mathbf{k}} \\ \Lambda^*_{\mathbf{k}} & J & \Lambda^*_{\mathbf{k}} & 0 \\ 0 & \Lambda_{\mathbf{k}} & J & \Lambda_{\mathbf{k}} \\ \Lambda^*_{\mathbf{k}} & 0 & \Lambda^*_{\mathbf{k}} & J \end{pmatrix}, \quad (2)$$

and $\Lambda_{\mathbf{k}} = J^A e^{-ik_x a/2} + J^{A'} e^{ik_x a/2} + J^B (e^{-ik_y b/2} + e^{ik_y b/2})$. The superscripts on each operator denote the two sublattices in the primitive unit cell. Quadratic terms from $\alpha = x, y, z$ are block-diagonalized into the same matrix, $\mathcal{M}_{\mathbf{k}}$, reflecting that each band is triply degenerate owing to a rotation symmetry. Dispersion relations are obtained by applying Bogoliubov transformation: by diagonalizing the matrix $\Sigma \mathcal{M}_{\mathbf{k}}$ ($\Sigma = \mathrm{diag}(1, 1, -1, -1)$), dispersion relations are obtained as

$$E_{\pm,\mathbf{k}} = \sqrt{J^2 \pm 2J|\Lambda_{\mathbf{k}}|^2}. \quad (3)$$

The observed triplon bands are well reproduced by the dispersion relation given by Eq. (3). The two bands with a large and small structure factor are represented by thick solid and thin dashed curves in Fig. 2a–c, respectively. The parameters $J$, $J^A$, $J^{A'}$, and $J^B$ are selected as 2.61, −0.24, −0.16, and −0.13 meV,

respectively, because these values best reproduce the observed dispersions. The simulated dispersion curves perfectly agree with the observed bands. These parameters are also consistent with $J = 2.4$ meV and $|J_p - J_d|$ (equals to $|J^A + J^{A'} + 2J^B|/2$) = 0.30 meV estimated from the magnetization curve[40]. This model is also supported by the energy slice presented in Fig. 4. The white solid curves in the figure indicate the region where triplon bands cross with a constant energy; an energy width of $\Delta E = \pm 0.10$ is taken into account from the energy window and energy resolution. They well describe the area where finite intensities are observed. Even around the gap energy, $2.60 \pm 0.06$ meV, weak intensities are detected because of the narrow band gap.

Interestingly, the gap between two triplet bands is topologically nontrivial. This can be easily understood by neglecting pair creation and annihilation terms in Eq. (1), which do not alter topological properties as we discuss later. Equation (1) is now reduced to a simple form:

$$\mathcal{H} \sim \sum_{\mathbf{k}} \left( t^{\dagger,1}_{+,\mathbf{k}} t^{\dagger,2}_{+,\mathbf{k}} \right) \mathcal{M}'_{\mathbf{k}} \begin{pmatrix} t^1_{+,\mathbf{k}} \\ t^2_{+,\mathbf{k}} \end{pmatrix}, \quad (4)$$

with a $2 \times 2$ matrix

$$\mathcal{M}'_{\mathbf{k}} = \begin{pmatrix} J & \Lambda_{\mathbf{k}} \\ \Lambda^*_{\mathbf{k}} & J \end{pmatrix} = J\mathbf{1} + \mathbf{d} \cdot \boldsymbol{\sigma}, \quad (5)$$

where $\mathbf{d}$ and $\boldsymbol{\sigma}$ represent a pseudomagnetic field, $\mathbf{d} = (\mathrm{Re}\Lambda_{\mathbf{k}}, -\mathrm{Im}\Lambda_{\mathbf{k}}, 0)$ and a Pauli matrix, respectively. The matrix leads to the eigenenergy $E_{\mathbf{k}} = J \pm |\mathbf{d}| = J \pm |\Lambda_{\mathbf{k}}|$. Thus, to open the energy gap between the two modes, a necessary condition is that $|\Lambda_{\mathbf{k}}| > 0$ for all $\mathbf{k}$, which requires interchain couplings to be weaker than the average of intrachain couplings, $|J^A + J^{A'}|/2 > |J^B|$, and bond alternation along each chain, $J^A \neq J^{A'}$.

The matrix $\mathcal{M}'_{\mathbf{k}}$ represents a 2D extension of the Su-Schrieffer-Heeger (SSH) model[42,43]. The SSH model describes electron motions in a 1D lattice with alternating hopping amplitudes and well demonstrates a topological distinction between nontrivial and trivial phases with respect to the number of edge states. The edge states in the SSH model exist at zero-energy because of a chiral symmetry. Even for bosonic systems such as triplons, the same topological distinction can be made between excited modes if an energy gap exists between them. In Ba$_2$CuSi$_2$O$_6$Cl$_2$, the hopping amplitudes of triplons are alternated along the $a$-direction but uniform along the $b$-direction, as shown in Fig. 1b. Thus, the interdimer network can be regarded as SSH chains along the $a$-axis coupled by interchain hoppings. Under $|J^A + J^{A'}|/2 > |J^B|$, the variation of $k_y$ only causes a small shift of $\mathbf{d}$ along $d_x$, keeping the winding number unchanged. Thus, the winding number can be defined for a fixed $k_y$, as that defined for the 1D system. The edge states are lifted up to energy $J$ because of the diagonal component in $\mathcal{M}'_{\mathbf{k}}$.

It should be noted that the alternating sequence of intrachain hopping amplitudes is opposite between nearest-neighbor SSH chains. The alternation yields two nontrivial gapped phases with changing intrachain hopping amplitudes[44], while the SSH chain yields one trivial and one nontrivial phases[42,43]. It is instructive to start from a coupled SSH chain, which consists of two SSH chains coupled with an opposite alternating sequence[44]. Its Hamiltonian is given by

$$\mathcal{M}^{\mathrm{SSH}}_{\mathbf{k}} = \begin{pmatrix} 0 & we^{-ik} + ve^{ik} + t \\ we^{ik} + ve^{-ik} + t & 0 \end{pmatrix}, \quad (6)$$

where $w$ and $v$ are intrachain hopping amplitudes, and $t$ is the interchain hopping amplitude connecting two chains. The nontrivial gapped phases appear in the weak interchain coupling region ($|v + w| > |t|$)[44]. The band topology of Ba$_2$CuSi$_2$O$_6$Cl$_2$ is

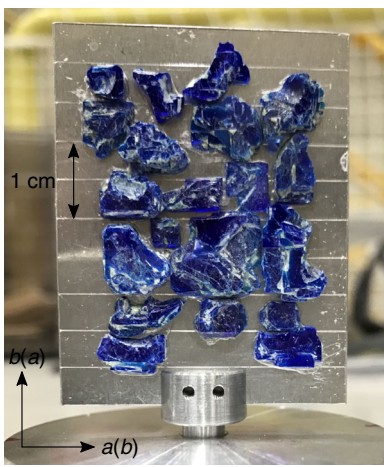

**Fig. 6** Aluminum plate. $Ba_2CuSi_2O_6Cl_2$ crystals co-aligned on an aluminum plate

almost the same as that of the coupled SSH chain. This can be understood by comparing nondiagonal components of the matrices in Eqs. (5) and (6): the only difference is the existence of dispersion along the $b$-axis. The winding number is evaluated by projecting a pseudomagnetic field $\mathbf{d}$ on $d_x$–$d_y$ space. Figure 5a depicts $d$ with exchange parameters set to those determined from the present experiment. For a fixed $k_y$, $\mathbf{d}$ represents a single ellipsis that surrounds the origin counterclockwise or clockwise; the winding number becomes $N = \pm 1$ for the present case. The two phases with the opposite winding number are separated by a phase boundary at $J^A = J^{A'}$, where the gap is closed. The winding number cannot be changed without closing the gap, because of $d_z = 0$.

The above discussion indicates that topologically protected edge states exist in the triplon band gap observed in $Ba_2CuSi_2O_6Cl_2$, as well as the SSH model. The symmetry-protected edge states cannot be removed by pair creation and annihilation terms, which can be confirmed by deriving the Berry connection. A Bogoliubov-de Gennes form of the two-sublattice triplon-band Hamiltonian is generalized into a $4 \times 4$ matrix as follows:

$$\mathcal{M}_{\mathbf{k}} = \begin{pmatrix} J\mathbf{1} + \mathbf{d} \cdot \boldsymbol{\sigma} & \mathbf{d} \cdot \boldsymbol{\sigma} \\ \mathbf{d} \cdot \boldsymbol{\sigma} & J\mathbf{1} + \mathbf{d} \cdot \boldsymbol{\sigma} \end{pmatrix}, \quad (7)$$

where $\mathbf{d} = (d_x, d_y, d_z)$ is a three-component real vector that is a function of $\mathbf{k}$. Then, by following the definition of the Berry connection for bosonic systems[13,18], its real part is obtained as

$$\mathrm{Re}A_{\pm,\mathbf{k}} = \pm \frac{1}{2|\mathbf{d}|(|\mathbf{d}| + d_z)} \left( d_x \frac{\partial d_y}{\partial k_x} - d_y \frac{\partial d_x}{\partial k_x} \right), \quad (8)$$

where $\pm$ represents two subbands. Irrespective of which gauge is selected, the real part of the Berry connection corresponds to that derived from a $2 \times 2$ matrix, $\mathcal{M}'_{\mathbf{k}} = J\mathbf{1} + \mathbf{d} \cdot \boldsymbol{\sigma}$, implying that topological properties are the same for both Hamiltonians, $\mathcal{M}_{\mathbf{k}}$ and $\mathcal{M}'_{\mathbf{k}}$ (see Supplementary Note 4 for a detailed derivation). In fact, for a 1D system in which $d$ is dependent on $k_x$, $d_z = 0$ corresponds to the Zak phase[45] $\gamma_{\pm} = -\int_{BZ} dk_x A_{\pm,\mathbf{k}}$ quantized into $\pm n\pi$, where $n$ corresponds to the winding number, $\int_{BZ} dk_x (d_x \partial d_y / \partial k_x - d_y \partial d_x / \partial k_x) / (2\pi |\mathbf{d}|^2)$. This indicates that edge states are protected by the equivalence between the two sublattices.

The presence of edge states is also confirmed by calculating the energy spectrum on chains with a finite length of 200 sites. As shown in Fig. 5b, two edge modes appear at the energy $J$ in addition to the bulk bands with dispersion relations described by Eq. (3). The flat dispersion with the band width of the order of

$10^{-11}$ meV induced by the hybridization between the edge modes (see Supplementary Fig. 3) reflects triplon densities localized at the edge: the alternation of hopping amplitudes induces one unpaired triplon at each edge, as illustrated in Fig. 5c. By reversing the magnitudes of $J^A$ and $J^{A'}$, the unpaired triplon density is also reversed between the two sublattices, which results in the opposite winding number.

It should be noted that the edge states in the present model are induced by a bipartite nature, and the edge states from the $S_z = 1$, 0 and $-1$ branches of triplet excitations are degenerate in the present model. The Dzyaloshinskii-Moriya interactions, which can lift the degeneracy, should be weaker than the energy resolution (0.1 meV) in $Ba_2CuSi_2O_6Cl_2$. This model is in contrast with the model based on $SrCu_2(BO_3)_2$ where interdimer Dzyaloshinskii-Moriya interactions and a magnetic field are crucial to produce the edge states[35,37]. The experimental detection of the edge states in the triplon band gap in $Ba_2CuSi_2O_6Cl_2$ is a future task.

In summary, triplet excitations in the dimerized quantum magnet $Ba_2CuSi_2O_6Cl_2$ were investigated via inelastic neutron scattering experiments. Two modes of the triplet excitations were detected together with a clear energy gap, which is induced by alternation of the interdimer interactions along the $a$-axis. This result is consistent with the newly determined crystal structure: the lack of $a$-glide allows interdimer interactions along the $a$-axis to alternate, while those along the $b$-axis become uniform. The whole dispersion relations are well reproduced with the three hopping constants $J^A$, $J^{A'}$, and $J^B$. The correspondence between the interdimer network of $Ba_2CuSi_2O_6Cl_2$ and a 2D extension of the coupled SSH model suggests the presence of the topological protected edge states in the triplon band gap.

## Methods

**Sample preparation.** Single crystals of $Ba_2CuSi_2O_6Cl_2$ were synthesized according to the previously reported procedure[40]. To synthesize single crystals of $Ba_2CuSi_2O_6Cl_2$, we first prepared $Ba_2CuTeO_6$ powder through a solid-state reaction. A mixture of $Ba_2CuTeO_6$ and $BaCl_2$ in a molar ratio of 1:10 was vacuum-sealed in a quartz tube, which acts as a $SiO_2$ source. The temperature at the center of the horizontal tube furnace was lowered from 1100 to 800 °C over 10 days. Plate-shaped blue single crystals with a maximum size of $10 \times 10 \times 1.5$ mm³ were obtained. The wide plane of the crystals was confirmed to be the crystallographic $ab$ plane by X-ray diffraction. The quartz tube frequently exploded during cooling to room temperature after the crystallization process from 1100 to 850 °C. To avoid hazardous conditions and damage to the furnace, a cylindrical nichrome protector was inserted in the furnace core tube.

**Single-crystal X-ray diffraction experiments.** Because the band gap of triplet excitations observed in $Ba_2CuSi_2O_6Cl_2$ cannot be described by the exchange model based on the original crystal structure[40], we reexamined the crystal structure at room temperature by using a RIGAKU R-AXIS RAPID three-circle X-ray diffractometer equipped with an imaging plate area detector. Monochromatic Mo-K$\alpha$ radiation with a wavelength of $\lambda = 0.71075$ Å was used as the X-ray source. Data integration and global-cell refinements were performed using data in the range of $3.119° < \theta < 30.508°$, and absorption correction based on face indexing and integration on a Gaussian grid was also performed. The total number of reflections observed was 73781, among which 5947 reflections were found to be independent and 5096 reflections were determined to satisfy the criterion $I > 2\sigma(I)$. Structural parameters were refined by the full-matrix least-squares method using SHELXL-97 software. The final R indices obtained for $I > 2\sigma(I)$ were $R = 0.0376$ and $wR = 0.0803$. The crystal data are listed in Supplementary Table 1. The structure of $Ba_2CuSi_2O_6Cl_2$ is orthorhombic $Cmc2_1$ with cell dimensions of $a = 13.9064(3)$ Å, $b = 13.8566(3)$ Å, $c = 19.5767(4)$ Å, and $Z = 16$. Its atomic coordinates and equivalent isotropic displacement parameters are shown in Supplementary Table 2.

**Inelastic neutron scattering experiments.** To explore the 2D nature of triplon excitations in $Ba_2CuSi_2O_6Cl_2$, its magnetic excitations were investigated using the cold-neutron disk chopper spectrometer AMATERAS installed in the Materials and Life Science Experimental Facility at J-PARC, Japan[46]. As shown in Fig. 6, twenty pieces of single crystals were coaligned on a rectangular Al plate so that an $a^*$ or $b^*$ direction for every crystal coincided with the edge directions of the Al plate. The Al plate was fixed in a vertical direction to set the $a^*$ and $c^*$ axes or $b^*$ and $c^*$ axes in the horizontal plane. Note that $a^*$ and $b^*$ axes cannot be distinguished

with each other because of crystallographic domains. Thus, both $a^*$ and $b^*$ components of a scattering vector are converted to a reciprocal lattice unit by the average of $a$ and $b$-axis lengths, which is 13.88 Å. The mixed domains do not matter in our analysis, because $a$ and $b$ axis lengths are almost the same, and triplet bands along $a^*$ and $b^*$ directions can be easily distinguished with each other, as described in the main text. Incident neutron energies were set to $E_i = (23.65, 5.924)$ meV and $(7.732, 3.135)$ meV by using repetition multiplication[47]. The coaligned crystals were rotated between a direction that forms a bond angle of $-35°$ and $55°$ with respect to the $c^*$ axis for $E_i = (23.65, 5.924)$ meV, while incident neutrons were kept parallel to the $c^*$ axis for $E_i = (7.732, 3.135)$ meV. The sample was cooled down to 0.3 and 2.5 K by using a $^3$He refrigerator. All the data collected were analyzed using the software suite UTSUSEMI[48].

## Data availability

Source data underlying Figs. 2a–c, f, 3a–d and 4a–c, Supplementary Figs. 1–3 are provided as a Source Data file. The data that support other findings of this study are available from the corresponding authors upon reasonable request.

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

## Acknowledgements

The neutron scattering experiment was performed under the J-PARC user program (Proposal No. 2016B0023). We express our sincere gratitude to M. Matsumoto, K. Nomura, K. Matan for useful discussions and comments. This work was supported by Grants-in-Aid for Scientific Research (A) (No. JP17H01142), (C) (No. JP16K05414 and JP17K05745), (Innovative Areas) (No. JP18H04504), Challenging Research (Exploratory) (No. JP17K18744) and Fund for the Promotion of Joint International Research (Fostering Joint International Research) (No. JP18KK0150) from the Japan Society for the Promotion of Science, and the CORE Laboratory Research Program "Dynamic Alliance for Open Innovation Bridging Human, Environment and Materials" of the Network Joint Research Center for Materials and Device".

## Author contributions

H.T. designed the experiment. K.T. and H.T. grew the crystal. K.T., K.Naw., N.K., H.T., S.O.-K. and K.Nak. performed the INS experiments. K. Naw. and T.J.S worked out the neutron-data and theoretical analysis. H.S., K.Naw., T.J.S., and H.U. performed the single crystal XRD experiments. K.Naw., T.J.S., and H.T. wrote the manuscript.

## Additional information

**Competing interests:** The authors declare no competing interests.

