## [Peer Review File · Nature Communications]

Reviewers' comments:

Reviewer #1 (Remarks to the Author):

The paper by Nawa et.al. studies triplon band structure of a two-dimensional antiferromagnetic magnet $\text{Ba}_2\text{CuSi}_2\text{O}_6\text{Cl}_2$. They observed a band splitting between two triplon bands. To understand this, they re-studied a crystal structure of $\text{Ba}_2\text{CuSi}_2\text{O}_6\text{Cl}_2$ and reveal that the material has a lower symmetry than the symmetry reported before. The author further clarified that the triplon band splitting can be understood as a triplon analog of SSH (Su-Schrieffer-Heeger) model. Apart from some awkward English expressions found in the paper, the scientific aspects of the paper seems to be sufficiently novel. I believe that the paper can be published in Nature communication, after the author reconsidered the comments below (optional).

1) In a model calculation given in Fig.5b, the authors show how the SSH end states can form a one-dimensional band along the b-axis. In a text, the authors also claims that 'The flat dispersion reflects triplon density localized at the edge. ...'. It seems for me that this complete flatness is an artifact of the oversimplified model, and the SSH end-state band could have a finite (presumably very small) dispersion along the b-axis. For example, Eq.(7) as well as Eq.(2) show that 'off-diagonal' two by two blocks (author called this as pair creation terms or pair annihilation terms in the text) contains a coupling between the same sublattices. This indicates that the SSH end state in one chain can hop to the SSH end state in its next-nearest-neighboring chains by using these pair creation terms. As Fig.5c suggests, this hopping would be presumably very small because two SSH end states in the next-nearest neighboring chains are always separated by a dimer in their common nearest-neighboring chain. Is the SSH end-state band completely flat in reality ?

2) In the text, the author mentioned about 'a chiral symmetry' (absence of dz term). For my understanding of the electronic SSH model, the chiral symmetry guarantees the presence of the SSH end state at $E=0$. In other words, the end state does exist at finite E, even when the system breaks this symmetry gradually by adding small dz term. In the electronic SSH model, the chiral symmetry is necessary for the presence of the zero-energy SSH end state, but not for the presence of SSH end state at finite E. Meanwhile, for quasi-particle boson system like the triplon systems considered in this paper, there is no much point of discussing about the 'zero-energy' SSH end state, because the end state of the triplon appears at finite energy E anyway. Certainly, the chiral symmetry (absence of dz term) would be useful for having the SSH end state of the triplon in a 'middle' of the triplon band gap. Nonetheless, since the middle of the band gap has no much meaning physically, I feels that there is no much point of emphasizing the 'chiral symmetry' in the present manuscript. In fact, Fig.1a and Fig.1c let me feel that the upper triplon band and the lower triplon band do not look like symmetric at all. Is 'chiral symmetry' important in this work ??

3) In Fig.5c, the authors show two pictures that correspond to $J^A > J^A'$ and $J^A < J^A'$. For me, both of them seem to support a SSH end state band along the b-axis like in Fig.5b. It is helpful, if the author can explicitly mention about the difference between these two cases in the viewpoint of a bundle of SSH end states in the present two dimensional system.

4) I feel that if J^B becomes larger, there may be a 'phase transition' from the viewpoint of the band topology of the two triplon bands. It is helpful if the author show a 'phase diagram' as a function of J^A and J^A' in the unit of J^B , where the authors could distinguish 'phases' in the diagram by different behaviour of the winding number as a function of ky.

5) Some English expressions and words in the manuscript sound weird for me. For example, in the second last sentence of the first paragraph, the text contains some word 'thermodynamic conductance'. What is it? Does author mean 'thermal conductance' by this?

6) Are two 4 by 4 matrices given in Eq.(7) and Eq.(2), $M^{(4)}_{\mathbf{k}}$ and $M_{\mathbf{k}}$ respectively, different? For me, they look identical. If so, why do the authors use the different symbols?

Reviewer #2 (Remarks to the Author):

In their manuscript, "Triplon band splitting and topologically protected edge states in the dimerized antiferromagnet" K. Nawa et al. give a detailed analysis of the triplet excitations of $\text{Ba}_2\text{CuSi}_2\text{O}_6\text{Cl}_2$. The Authors first establish the crystal structure using XRD experiments on single-crystals. This revealed the lowering of previously established symmetry and the lack of the a-glide which allows the alternation of exchange coupling along the a-axis. As a consequence, the effective model describing the triplet dynamics contains alternating hopping amplitudes thereby realizing the Su-Schrieffer-Heeger (SSH) model in a quantum magnet.

After establishing the new symmetry of $\text{Ba}_2\text{CuSi}_2\text{O}_6\text{Cl}_2$, the very reason for the triplon bands to become topologically nontrivial, the Authors construct a symmetry-allowed Hamiltonian and calculate the triplet excitations for various directions in momentum space. The theoretical model matches well the measured neutron spectra and provides an easy to understand interpretation for the triplet bands. The Authors find that $\text{Ba}_2\text{CuSi}_2\text{O}_6\text{Cl}_2$ can be regarded as one-dimensional SSH models coupled along the (uniform) b-direction. The topological invariant, the Zak phase (and winding number of the pseudo-magnetic field) is computed showing that the triplets are indeed nontrivial. To complete the study, a finite geometry is considered to reveal the localized edge modes coming from the unpaired triplons at the ends of the SSH chains.

To my best knowledge, this is the first physical realization of the SSH model in real material, other experimental realizations were only achieved using ultracold atoms. Therefore, I think this work is important and expectedly will inspire further theoretical and experimental studies. The manuscript is well written and the results are explained carefully with sufficient details for reproducing the findings. Therefore, I recommend publication of this paper in Nature Communications without reservation.

Nevertheless, I have a few minor suggestions/questions that may help readers.

1. Throughout the paper the alternation of the exchange/hopping is considered along the a-axis, a choice also followed in the theoretical model. The measured neutron spectrum, however, reveals that there are two domains corresponding to modulation along the a or b axes. This is the reason why both of the theoretically calculated dispersions, the one along H and that along K, are visible in the experiment. It would probably help potential readers to emphasize the presence of domains more as the reason for the presence of two bands.

2. The localized edge modes are obtained in the theoretical model due to the unpaired triplets at the two ends of the finite sample as illustrated in Figure 5c. As we learned from the neutron experiments, there are two domains present. Would we also find localized unpaired triplets at the boundary of domains too? Are there other kinds of edge states coming from trimers like in the case of the original SSH model?

A very minor comment: The Authors say that the edge modes (all three corresponding to the three kinds of triplets) are two-fold degenerate. Should there be a tiny splitting due to hybridization of the two edges?

3. All the modes, including the two edge modes, have an additional three-fold degeneracy due to the spin degree of freedom. What are the ramifications of having an extra spin-1 degree of freedom in comparison with the conventional SSH model? Are there any?

- Judit Romhányi

Comment from the first referee

1) In a model calculation given in Fig.5b, the authors show how the SHH end states can form a one-dimensional band along the b-axis. In a text, the authors also claims that 'The flat dispersion reflects triplon density localized at the edge. ...'. It seems for me that this complete flatness is an artifact of the oversimplified model, and the SSH end-state band could have a finite (presumably very small) dispersion along the b-axis. For example, Eq.(7) as well as Eq.(2) show that 'off-diagonal' two by two blocks (author called this as pair creation terms or pair annihilation terms in the text) contains a coupling between the same sublattices. This indicates that the SSH end state in one chain can hop to the SSH end state in its next-nearest-neighboring chains by using these pair creation terms. As Fig.5c suggests, this hopping would be presumably very small because two SSH end states in the next-nearest neighboring chains are always separated by a dimer in their common nearest-neighboring chain. Is the SSH end-state band completely flat in reality?

The SSH edge states are not completely flat and have a very weak dispersion. However, this is not caused by pair creation and annihilation terms but by the hybridization of two (almost-degenerate) edges modes. In fact, the dispersion does not depend on whether pair creation and annihilation terms are present or not but strongly depend on the number of sites ($2 \cdot N_a$) along the chain, as shown in the next figure. The width of the band dispersion exponentially decreases as N_a increases, and becomes completely zero in the large N_a limit. Since the term "flat" is not accurate, we comment that very weak dispersion is present in line 205, and add a figure in extended data (Extended figure 4) to show that the edge modes are not affected by the pair creation and annihilation terms.

Figure. Dispersion of the edge modes under 100 (left) and 200 sites (right) along the chain ($N_a = 50$ and 100 , respectively). Red crosses and black circles indicate the modes derived from Hamiltonian where pair creation and annihilation terms are present and absent, respectively.

2) In the text, the author mentioned about 'a chiral symmetry' (absence of dz term). For my understanding of the electronic SSH model, the chiral symmetry guarantees the presence of the SSH end state at $E=0$. In other words, the end state does exist at finite E , even when the system breaks this symmetry gradually by

adding small dz term. In the electronic SSH model, the chiral symmetry is necessary for the presence of the zero-energy SSH end state, but not for the presence of SSH end state at finite E . Meanwhile, for quasi-particle boson system like the triplon systems considered in this paper, there is no much point of discussing about the 'zero-energy' SSH end state, because the end state of the triplon appears at finite energy E anyway. Certainly, the chiral symmetry (absence of dz term) would be useful for having the SSH end state of the triplon in a 'middle' of the triplon band gap. Nonetheless, since the middle of the band gap has no much meaning physically, I feel that there is no much point of emphasizing the 'chiral symmetry' in the present manuscript. In fact, Fig.1a and Fig.1c let me feel that the upper triplon band and the lower triplon band do not look like symmetric at all. Is 'chiral symmetry' important in this work ??

We agree that a terminology "chiral symmetry" is not used correctly in our manuscript. As the referee indicates, our model do not meet requirements for the chiral symmetry. Although energy spectrum should become *symmetric* with respect to *zero energy* if a chiral symmetry is preserved, the energy spectrum appears *asymmetric* with respect to *energy J* in our model. Thus, we removed the terminology "chiral symmetry" that is used to explain our model. On the other hand, we still think it is better to refer to a chiral symmetry in the SSH model to indicate the difference between the SSH model and our model. Thus, we add explicit explanations that the SSH model has zero-energy modes because of the chiral symmetry in line 177 but the edge states are lifted up to energy J in our model in line 183. Note that even the bulk energy spectrum is clearly asymmetric, edge modes remain symmetric with respect to J , as shown in the above figure. This point is also indicated in the last part of the extended data.

3) In Fig.5c, the authors show two pictures that correspond to $J^A > J^{A'}$ and $J^A < J^{A'}$. For me, both of them seem to support a SSH end state band along the b -axis like in Fig.5b. It is helpful, if the author can explicitly mention about the difference between these two cases in the viewpoint of a bundle of SSH end states in the present two dimensional system.

We slightly revised the sentence at line 218 so that readers can understand that the edge triplon densities are reversed between the two sublattice.

4) I feel that if J^B becomes larger, there may be a 'phase transition' from the viewpoint of the band topology of the two triplon bands. It is helpful if the author show a 'phase diagram' as a function of J^A and $J^{A'}$ in the unit of J^B , where the authors could distinguish 'phases' in the diagram by different behavior of the winding number as a function of k_y .

As shown in Figure 5c, our model can be regarded as a bundle of a coupled SSH chains, and thus, the phase diagram is almost the same as that of a single coupled SSH chain presented in the reference 44. From this viewpoint, we think that the phase diagram is unnecessary and explanations added in the line 172-173 and 192 are enough for readers to understand the difference.

5) Some English expressions and words in the manuscript sound weird for me. For example, in the second last sentence of the first paragraph, the text contains some word 'thermodynamic conductance'. What is it ? Does author mean 'thermal conductance' by this ?

Thanks for indicating our awkward expressions. "Thermodynamic conductance" is modified into "thermal conductance". We also revised English expressions in the whole manuscript for better understanding.

6) Are two 4 by 4 matrices given in Eq.(7) and Eq.(2), $M^{(4)}_k$ and M_k , respectively, different ? For me, they look identical. If so, why do the authors use the different symbols?

$M^{(4)}_k$ and $M^{(2)}_k$ are modified into M_k and M'_k , respectively, so that readers can understand that they are identical in case of $d = (\text{Re}\Lambda_k, -\text{Im}\Lambda_k, 0)$.

Comment from the second referee

1. Throughout the paper the alternation of the exchange/hopping is considered along the a-axis, a choice also followed in the theoretical model. The measured neutron spectrum, however, reveals that there are two domains corresponding to modulation along the a or b axes. This is the reason why both of the theoretically calculated dispersions, the one along H and that along K, are visible in the experiment. It would probably help potential readers to emphasize the presence of domains more as the reason for the presence of two bands.

Thanks for the recommendation. The domain for each piece of crystals does not mean that the modulation appears both along the a or b axes but just indicate that the a-axis of a single domain and the b-axis of another domain coexist along the same edge. We revised the explanation at line 85-88.

2. The localized edge modes are obtained in the theoretical model due to the unpaired triplets at the two ends of the finite sample as illustrated in Figure 5c. As we learned from the neutron experiments, there are two domains present. Would we also find localized unpaired triplets at the boundary of domains too? Are there other kinds of edge states coming from trimers like in the case of the original SSH model?

A very minor comment: The Authors say that the edge modes (all three corresponding to the three kinds of triplets) are two-fold degenerate. Should there be a tiny splitting due to hybridization of the two edges?

Answer to the first question:

As we can deduce from a schematic view similar to that illustrated in Figure 5, localized unpaired triplets can be also present at the boundary of domains. Excitations of the localized unpaired triplets may appear at different energy, depending on the inter-domain interactions.

Answer to the second question:

Other kinds of edge states such as zigzag edge states can be illustrated as follows. Note that the band topology becomes different from that discussed in the present manuscript since the definition of the unit cell should be also changed.

Answer to the third question:

Yes, there is a tiny splitting that is an order of $10^{-(N_a/10 + 1)}$. We add the figure representing the weak dispersion (Extended Figure 4) and the detailed explanations to avoid confusion.

3. All the modes, including the two edge modes, have an additional three-fold degeneracy due to the spin degree of freedom. What are the ramifications of having an extra spin-1 degree of freedom in comparison with the conventional SSH model? Are there any?

Since the chemical potential of triplons can be controlled by applying a pressure or magnetic field, we may investigate the effect of quantum fluctuations towards the edge states by tuning the system near a quantum critical point. Since quantum fluctuations cannot be introduced in conventional electron systems such as those in the SSH model, investigating such edge states under quantum fluctuations would be impactful.